# Scenario Analysis of Initial Water-Rights Allocation to Improve Regional Water Productivities

**Tienan Li [1], Xueting Zeng [2,*], Cong Chen [3], Xiangmin Kong [4], Junlong Zhang [5], Ying Zhu [6], Fan Zhang [2] and He Dong [1]**

1 Institute of River and Lake Environment, Heilongjiang Provincial Hydraulic Research Institute, Harbin 150080, China; litienan0019@163.com (T.L.); albert121224@163.com (H.D.)
2 School of Labor of Economics, Capital University of Economics and Business, Beijing 100070, China; 22018205004@cueb.edu.cn
3 Donlinks School of Economics and Management, University of Science and Technology Beijing, Beijing 100083, China; chencong@ustb.edu.cn
4 School of Fundamental Science, Beijing Polytechnic, Beijing 100176, China; Xiangming0913@126.com
5 College of Environment Science and Engineering, University of Qingdao, Qingdao 266071, China; zjunlong0801@163.com
6 School of Environment and Municipal Engineering, Xi'an University of Architecture and Technology, Xi'an 710055, China; zhuyingxauat@163.com
* Correspondence: zengxueting@cueb.edu.cn; Tel.: +86-0108-395-1881

**Abstract:** In this study, an initial water-rights allocation (IWRA) model is proposed for adjusting the traditional initial water-rights empowerment model based on previous water intake permits, with the aim of improving the productivity of water resources under population growth and economic development. A stochastic scenario with Laplace criterion mixed fuzzy programming (SSLF) is developed into an IWRA model to deal with multiple uncertainties and complexities, which includes dynamic water demand, changing water policy, adjusted tradable water rights, the precise risk attitude of policymakers, development of the economy, and their interactions. SSLF not only deals with fuzziness in probability distributions with high satisfaction degrees, but also reflects the risk attitudes of policymakers with the Laplace criterion, which can handle the probability of scenario occurrence under the supposition of no data available. The developed IWRA model with the SSLF method is applied to a practical case in an alpine region of China. The results of adjusted initial water rights, optimal water-right allocation, changed industrial structure, and system benefits under various scenarios associated with risk attitudes and water productivity improvement were obtained and analyzed. It was found that the current initial water-rights allocation scheme based on previous intake water permits is not efficient, and this can be modified by the IWRA model. Based on the strategies of drinking safety and ecological security, the main tradeoff between agricultural and industrial water rights can facilitate optimization of the current initial water-rights allocation. This can assist policymakers in producing an effective plan to promote water productivity and water resource management in a robust and reliable manner.

**Keywords:** water rights; stochastic scenario analysis; fuzzy credibility programming; optimization; alpine region

---

## 1. Introduction

The increased stress of water scarcity is bringing about the need for more efficient water resource management, such as legal foundations, water price, resource protection policies, technology improvement, and water allocating schemes [1]. Particularly in the context of climate change,

the over-allocation and inefficient water use due to unclear targets, weak enforcement, and limited stakeholder involvement can bring about low efficiencies in water allocations [2]. For example, Chen et al. [3] developed an APSIM model to achieve a more efficient and sustainable utilization of limited water resources, which can improve understanding of how crop productivity and water balance components respond to climate variations. Pereira et al. [4] incorporated water use concepts and performance descriptors into a new indicator framework to support water conservation and water saving, with the aim of improving the overall performance and productivity of water use. However, the above methods lack an incentive for promoting water productivity in the long run. Therefore, water rights trading can be introduced as an incentive to improve water productivity by encouraging water flow from low to high values (i.e., endogenous impetus) by a market approach in the long term; this is already utilized in a number of countries worldwide [5,6]. China has carried out a water rights trading project within the water resource management system in recent years, with the aim of addressing a water crisis [7]. However, the implementation of water-rights trading requires the empowerment of initial water rights (i.e., the water rights weighted by the government to water users, firstly for future buying or selling activities through the water market) and a perfect market, neither of which have been constructed perfectively in China today. Therefore, the government of China has designed a new initial water-rights allocation (IWRA) scheme for future water-trading implementation. At the beginning of the initial water-rights empowerment from 2016, the initial water rights were estimated according to previous water intake permits that has been assigned twenty years ago, which means they do not cover increased water demand due to the expanded population, increased irrigation, and accelerated industrial development. For example, some water users (e.g., companies) have been closed or their production reduced since the original permits were issued, however can obtain the same water rights as their previous water intake permit (assigned 20 years ago), leading to over-allocation. Meanwhile, some water users that are not consistent with local economic development (maybe with lower water efficiency and higher pollution levels) may also obtain their initial water rights due to the unchangeable water intake permit, resulting in low productivity and efficiency ("efficiency" is denoted as a volume of water rights that could benefit from production). The above problems can result in the expected water demand exceeding the limited available water rights, a problem which requires a different, comprehensive plan for initial water-rights allocation (IWRA).

In alpine regions of China such as Heilongjiang River Basin, the government carried out water rights reform in which the first step was initial water-rights empowerment, which requires a comprehensive IWRA scheme to address complex natural and artificial situations. For instance, the natural characters of irregular rainfall, long frost periods, and low temperature can result in an uneven spatial and temporal distribution of water resources, which may bring about severe seasonal water scarcity that reduces the total available water rights randomly. Meanwhile, the national Northeast revitalization policy and productive crop protection strategy have accelerated the water demand. The initial water-rights allocation model, based on previous water intake permits, would enhance the conflicts between water demand planning, water-rights allocation, and water resource management. Therefore, a more comprehensive water-rights allocation plan or model to address the challenges of accelerated economic development, increased water demand, random water shortages, unreliable water supply, and low water productivity is required.

Previously, various research works on water-rights allocation models have been undertaken. For instance, Kreutzwiser et al. [2] assessed the ability of the Permit to Take Water program to identify opportunities to enhance water-rights allocation, where various alternatives and the amount of water withdrawn associated with municipal planning policies, stakeholder input, and clear and legally-established water use priorities have been analyzed. Yusuke and Nicholas [8] analyzed intra- and inter-state conflicts and corresponding changes in water-rights management policies using a population data set of irrigation wells in the Republican River Basin. Bof et al. [9] designed various scenarios for allocating water rights in the Paracatu River Basin, Brazil, to improve water management and minimize the risk of water deficits. Latinopoulos and Sartzetakis [10] developed a hybrid discrete

and continuous time model to reflect farmers' myopic behaviors and water management to optimize groundwater usage in irrigation, and allowing tradable water rights. The above research works have shown that a water-rights allocation can be influenced by a number of complex factors, such as myopic behavior, stakeholder input, legal establishments, and intra- and inter-state conflicts, which should be incorporated into a framework to reflect the combined effect on the outcome. Thus, a scenario analysis (SA) to reflect complex influence impacts within a system under a set of 'possibility space' futures is utilized. In an SA, plenty of impact factors can be considered in the scenario assumption; then, a "possibility space" can be explored to approach imprecise information associated with the interactions between many impact elements and decision outcomes [11–13]). Formerly, a number of researchers have considered the use of scenario analysis to reflect the complex relationship of impact factors in water-rights management. For example, Veettil et al. [14] used a discrete choice model to analyze various scenarios associated with local irrigation water governance, four types of water pricing methods, and different water-rights situations (to generate the optimal strategy), which can be accepted by farmers and the government. Zeng et al. [7] proposed an exact scenario analysis method to reflect various system benefits under changed water-rights scenarios, which could support adjustment to the current water trading policy in an arid region of China. Wang et al. [15] developed a scenario analysis modeling framework and a mixed integer optimization model (MIOM) to optimize water intake on/off events, which could improve the efficiency of water-rights management in comparable basins. In SA, the risk attitudes (risk seeking/risk neutral/risk avoiding) of policymakers are deemed an important factor in the process of scenario design; various subjective estimations of policymakers (including risk attitudes) to natural features and artificial features can influence decision-making, leading to varied outcomes [11]. For instance, meteorological variations such as rainfall and their frequencies of peaks are deemed as stochastic factors in water-rights management, which may influence the total available water rights. In the context of climate change, policymakers with a risk seeking attitude would empower more water availability through available water rights for trading, leaving less water resources as reserved water for emergencies. In contrast, policymakers with a risk avoiding attitude would reserve more water resources for emergencies. However, the probability of scenario occurrences regarding to varied risk attitudes is random, which would be influenced by the private experiences and personality traits of policymakers. Thus, stochastic programming (SP) is introduced into SA, which can express the probability of a scenario as a probabilistic distribution [16,17]. In a stochastic scenario analysis (SSA), random political considerations (such as food safety policy, industry adjustment strategy, and environmental protection regulation) and subjective estimation due to risk attitudes can be incorporated into scenario design as probabilistic distributions. However, in a practical IWRA, the data of the risk attitudes of policymakers is limited and can be difficult to obtain adequately. In this instance, Laplace's criterion can be introduced into SSA to handle the probability of scenario occurrence under the supposition of no data available, where the probability of scenario occurrence can be supposed to be reasonably equal [13,18].

Nevertheless, there is fuzzy information in IWRA which cannot be handled by a stochastic scenario analysis with Laplace's criterion (SSL). For instance, SSL cannot handle fuzziness in economic data associated with water efficiency due to data deficits or estimative error. Thus, fuzzy programming (FP) is introduced to reflect fuzziness in the observed information, to improve the vagueness expression in goals or constraints caused by inartificial factors [19]. For instance, inputs such as losses due to a water-rights deficit are hard to calculate precisely; meanwhile, the productivity of water rights is difficult to obtain as an accurate value in most cases. Therefore, a credibility measure is adopted into FP to deal with the possibility and necessity degrees of event occurrence, which can generate flexible results with high satisfaction degrees [20–22]. On the whole, various types of uncertain information and corresponding interactions can fortify the complexity of IWRA. Unfortunately, few research works have yet focused on coupling different methods (e.g., SA, SP, FP and Laplace criterion) into a framework.

Therefore, the objective of this study is to build an initial water-rights allocation (IWRA) model for adjusting the traditional initial water-rights modes based on previous water intake permits, with the aim of improving the productivity of water resources under population growth and economic development. A stochastic scenario-based with Laplace criterion mixed fuzzy programming (SSLF) is developed to deal with multiple uncertainties in the IWRA model. SSLF can not only handle uncertainties expressed as probability distributions, but also reflect the occurrence of a scenario into a future space. The proposed SSLF method was used with the IWRA model on a real case in Heilongjiang province, a typical alpine region in China. The results of water rights being withdrawn, optimal initial water rights, industrial structure, and system benefits under various scenarios were analyzed. This allowed us to identify the optimized irrigative scale, industrial layout, and water productivities and resources management policies based on risk analysis. It could also support policymakers in adjusting the current views of economic development and water-rights allocation schemes in a robust manner.

## 2. Materials and Methods

### 2.1. Overview of the Study Region

Hulin county is located in the east of Heilongjiang province, between north latitude 45°23′ to 46°36′, east longitude 132°11′ to 133°56′ (as shown in Figure 1). It has a total area of 9330 square kilometers [23]. Hulin is located in a cold temperate continental monsoon area, which belongs to the Sanjiang plain moderate humid climate zone. In this climate zone, winters are long and cold; summers are short, warm and rainy. The annual average temperature is 3.5 °C; meanwhile, the average annual evaporation and average rainfall are 1110.7 mm and 566.2 mm. Precipitation is mainly concentrated in June, July and August, accounting for 53% of the total precipitation in 2012 [24]. The average annual relative humidity is 70%. The snow melts in late February and freezes for about 180 days. There are 10 branches of the Wusuli River, including the Muling River, Qihulin River, Abqin River, Songacha River, Dumu River, Xiaomu River and Qili qinhe River in Hulin county. The average annual water resources of the city are 1.948 billion cubic meters, including 720 million cubic meters of groundwater resources and 533 million cubic meters of recoverable water [25]. The annual average surface water resources comprise 1.48 billion cubic meters. The abundant water resources are very useful for developing regional agriculture (such as irrigation), which is considered the pillar industry in Heilongjiang province.

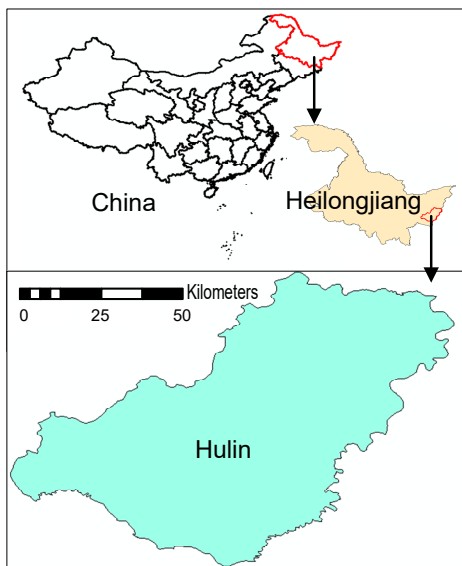

**Figure 1.** Study area.

The Hulin county has one sub-district office, seven towns and four townships, two communities, and 85 villages under its jurisdiction. In 2017, the GDP of Hulin county reached 13.84 billion yuan,

up 7 percent year-on-year [26]. The added value of primary, secondary and tertiary industries increases by 7%, 6.8% and 7.3%, respectively. The structure of green agriculture is growing too. A total of 270,000 ha of high-efficiency crops have been developed, accounting for more than 10% for the first time [26]. Meanwhile, based on the quantity and quality of the industrial economy increase, the added value of industrial enterprises reached 780 million yuan.

Figure 2 presents the framework of an initial water-rights allocation (IWRA) model in an alpine region. In the study region, water demand has shown an increasing trend due to accelerated economic development, which may exceed what the natural system can supply. Thus, Hulin has confronted the challenges as follows: (a) Hulin county is one of the most important bases of grain production in Heilongjiang province, and requires more stable water resources to improve its agricultural development. Increasing water demand for agriculture may reach the high-point of what the natural system can maintain. Thus, an effective system, such as water trading, should be considered in any water allocation mode. (b) Although water trading is an effective way to remit the conflict between the changing industrial structure and an obsolete water allocation mode, it requires a perfectly contestable market and available water rights. Thus, a comprehensive initial water-rights empowerment (or allocation) should be the first step of water trading. (c) In order to achieve initial water-rights allocation (i.e., IWRA) quickly, initial water rights are calculated by the previous water intake permits, however this method is not suitable for current economic development. Thus, optimization of initial water-rights allocation is desired in a water trading scheme. For instance, some industrial enterprises or agricultural plants that have been closed down now obtained water intake permits twenty years ago; they can obtain the same initial water rights for sale to get benefits. This could result in lower productivities of the water resources. Thus, a rational and optimal IWRA model is required to reallocate initial water rights. (d) Accelerated industrialization and excessive agricultural exploitation would bring about great stresses on water resources and the environment, and could increase the occurrences of disastrous extreme events (such as water resource pollution, soil erosion and water deficits), resulting in economic losses. Thus, coordination of the water relationship among industry, agriculture, municipalities, and ecology is another important issue in an IWRA. (e) There is a lot of uncertain information in an IWRA scheme due to socioeconomic and climatic changes, suggesting the need for a comprehensive plan to improve the accuracy of water resource management.

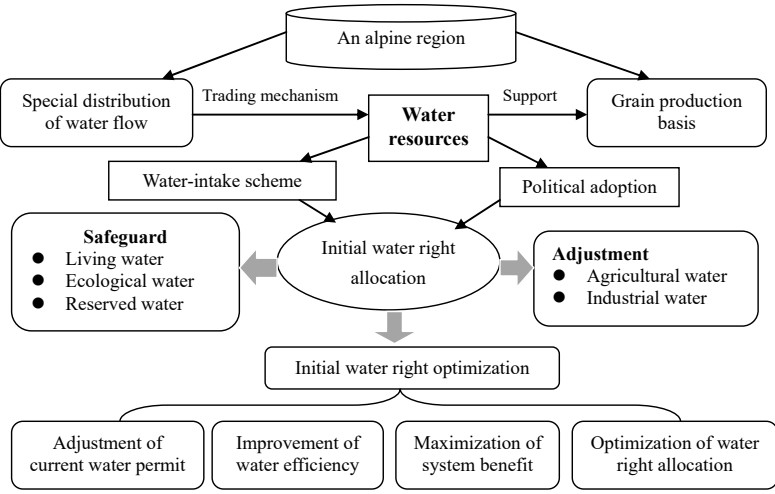

**Figure 2.** Framework of an initial water-rights allocation (IWRA) model in an alpine region.

## 2.2. Development of SSLF in an IWRA Model

Figure 3 presents the framework of the development of SSLF in an IWRA model. Since a number of objective and subjective factors can result in varied types of uncertainties, the complexity of the IWRA is increased [27]. Therefore, these uncertainties can be considered in an IWRA model to address

challenges due to population growth and economic development. For example, plenty of impact factors can be deemed as inputs, which may generate a "possibility space" that can be explored to approach imprecise information associated with the interactions between many impact elements and decision outcomes [13]. Therefore, a scenario analysis (SA) method can be used to simplify complex management into various scenarios that can reflect interactions between complex factors and decision outcomes [11]. Meanwhile, stochastic programming can be adopted into SA to reflect the probability of scenarios as probabilistic distributions [12]. In general, among the various impact factors, the risk attitude of a policymaker is an important parameter to influence decision-making, resulting in political changes [28]. However, various risk attitudes such as risk seeking, avoiding, and neutral attitudes in random scenario are difficult to calculate precisely. Therefore, Laplace's criterion (LC) is adopted into SA for handling uncertain probabilities of scenario occurrence due to limited data availability, based on supposition of equal probabilities of scenario occurrence if the sample size approaches infinity [13,18]. This stochastic scenario analysis, undertaken using the Laplace's criterion (SSL) method, can help policymakers to improve the balance of the relationship between the expected input performances and allow them to choose alternatives with maximum values. However, in an IWRA model, some parameters (such as limited economic data or meteorological data) in the right- and left-hand sides of objective functions and constraints are expressed as vagueness due to limited data and estimative error, which cannot be tackled by SSL. Hence, a type of fuzzy programming called fuzzy credibility constrained programming (FCP) can be added to reflect fuzzy information regarded as a possibility distribution, which can express the relationship between satisfaction degree and system-failure risk [19]. A stochastic scenario-based with Laplace criterion mixed fuzzy programming (SSLF) can be formulated, for which the detailed solution is shown in Appendix A.

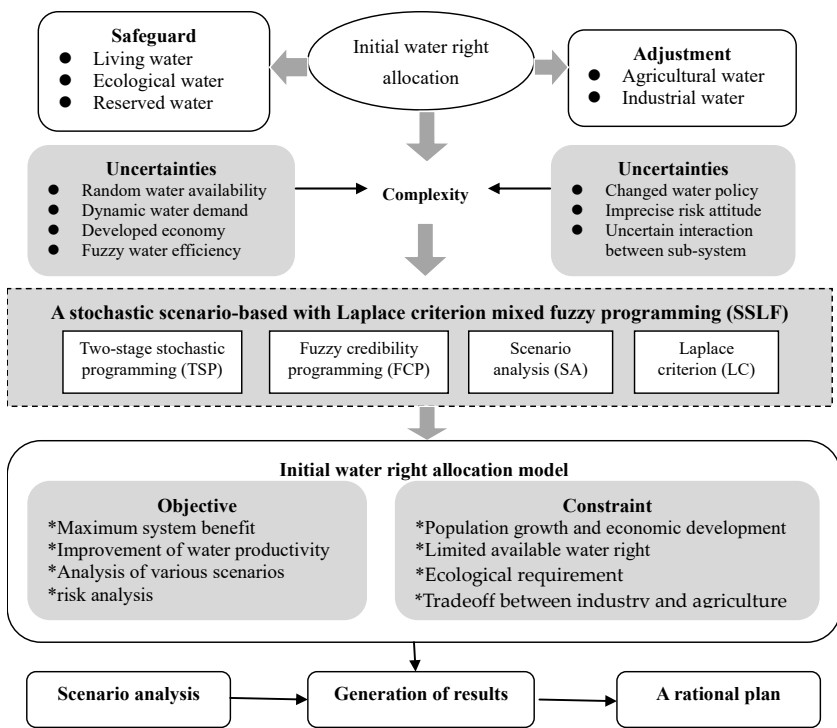

**Figure 3.** Framework of development of SSLF in an IWRA issue.

The developed SSLF can deal with uncertainties expressed as probability distributions, but also reflects the occurrence of a scenario into a future space. In summary, the solution process of SSLF in an IWRA model can be summarized as follows:

Step 1: Analyzing objective and subjective impact factors in an IWRA model. It can be deemed as an input of the SA method (including scenario assumption and analysis) to produce a "possibility space".

Step 2: Assumption of the probability of a scenario is random in SA, which can be expressed as the probability distribution (i.e., $\sum\limits_{k=1}^{K} P_k$).

Step 3: Adopting Laplace's criterion (LC) into SA to suppose the probabilities of scenario occurrence equally if the sample size approaches infinity (i.e., $\sum\limits_{k=1}^{K} P_k = 1/K$).

Step 4: Introducing $\lambda$ level (credibility measure) to reflect risk violations and confidence degrees when the information is fuzzy on both sides of the constraints.

Step 5: Based on the conception of the credibility measure $\lambda$, the credibility measure can be proven $\lambda > 0.5$, then the $\lambda$ level can be substituted into the constraint (i.e., $c_n^2 + (1 - 2\lambda)(c_n^2 - c_n^1), n = 1, 2, \ldots, N_1$).

Step 6: Obtaining an optimal solution of SSLF.

Step 7: Taking SSLF into an IWRA model in an alpine area.

In this study, the water resources management department can be deemed as a water manager to replace the traditional initial water-rights allocation mode, in which water is allotted according to water intake permits issued twenty years ago (the water intake permit equals initial water rights), with the aim of maximizing the system benefit in an IWRA system. In fact, the traditional water rights allocation mode is a backward allocation mode with lower productivity, since it cannot consider socioeconomic development and population growth. Under these situations, although a number of water users have stopped production, they can obtain water rights based on their previous water intake permit despite this leading to lower or even no productivity. However, due to population growth and economic development, the available water rights cannot satisfy the increase to the initial water rights demand. Thus, the water manager is responsible for reallocating water rights to improve the productivity of water resources. At the beginning of the year, the water manager can obtain the expected target from various water users based on the average demand for the most recent 3 years, with consideration of the regional development plan. They can allocate initial water rights to various sectors, including municipal, agricultural, industrial and ecological users, based on previous water intake permits (from 20 years ago) at first. Then, the water manager can reduce irrational initial water-rights allocation for water users that have closed or have lower productivities, and this can be further adjusted once every three years. Since the risk attitude can reflect the decision process of the water manager, the Laplace criterion can be introduced into the IWRA model as follows:

$$
\begin{aligned}
&\max Outcome_{Laplace}(A_{ih}) \\
&= \frac{1}{K} \times \left\{ \max_{d \in D} S_{input} \left[ \left( \begin{matrix} rsa_{i1}^1 & rsa_{i2}^1 & \ldots & rsa_{ih}^1 \\ rsa_{i1}^2 & rsa_{i2}^2 & \ldots & rsa_{ih}^2 \\ \ldots & \ldots & \ldots & \ldots \\ rsa_{i1}^d & rsa_{i2}^d & \ldots & rsa_{ih}^d \end{matrix} \right) + \left( \begin{matrix} raa_{i1}^1 & raa_{i2}^1 & \ldots & raa_{ih}^1 \\ raa_{i1}^2 & raa_{i2}^2 & \ldots & raa_{ih}^2 \\ \ldots & \ldots & \ldots & \ldots \\ raa_{i1}^d & raa_{i2}^d & \ldots & raa_{ih}^d \end{matrix} \right) \right] \times (a_{ih}^1, a_{ih}^2, \ldots, a_{ih}^d) \right\}* \\
&\left\{ \left[ \left( \sum\limits_{m=1}^{20} \sum\limits_{s=1}^{2} BIC_{ms} \times wic_{ms} \times (1 + \alpha) - \sum\limits_{m=1}^{20} \sum\limits_{s=1}^{2} CIC_{ms} \times Yic_{ms} \right) - \sum\limits_{m=1}^{20} TIC_{ms} \times wic_{ms} \times \alpha \right] + \\
&\left[ \left( \sum\limits_{n=1}^{30} \sum\limits_{s=1}^{2} BAC_{ns} \times wac_{ns} \times (1 + \beta) - \sum\limits_{n=1}^{30} \sum\limits_{s=1}^{2} CAC_{ns} \times Yac_{ns} \right) - \sum\limits_{n=1}^{30} \sum\limits_{s=1}^{2} TAC_{ns} \times wac_{ns} \times \beta \right] \\
&+ \left[ BMC_j \times wmp_j \times (1 + \eta) - TMC_j \times wmp_j \times \eta \right] + BEC_i \times wep_i \right\}
\end{aligned}
\tag{1}
$$

The explanations of the variables and parameters are in Appendix B. Due to implementation of drinking safety, municipal water rights are allocated based on the domestic water quota for rural and urban residents, which are 60 and 80 L/(person·d). Meanwhile, ecological security has been advocated for over the years, which has resulted in ecological water use being safeguarded for environmental sanitation in the study region. Therefore, the tradeoff between agricultural and industrial users has facilitated optimized water rights results based on Model (1), where m is water users (i.e., enterprises in the industrial sector) in various industries in Hulin county; n is various irrigation districts in agricultural sectors; and s is the water sources, including surface water and underground water. Since the current water rights plan is irrational (i.e., calculated by previous intake water permits), we have to adjust the current initial water rights based on actual productivities of water resources.

This means that the irrational initial water rights should be reduced or withdrawn from original users. Under this situation, $Yic_{ms}$ and $Yac_{ns}$ are the adjusted water rights. $CIC_{ms}$ and $CAC_{ns}$ are economic losses due to insufficient water rights for a user after water-rights reduction/withdrawal. Moreover, the constraints can be formulated as follows:

(1) Constraint of available water resources: Model (2) displays the balance among initial water-rights empowerment based on previous water intake permits, reduced water rights, and available water resources, where $q$ is the total water availability in the study region, which is influenced by regional climatic features (such as the rainfall situation). Since it is hard to calculate a precise value due to data deficits, it can be expressed as a fuzzy sect. Therefore, fuzzy credibility programming is adopted for reflecting this vagueness:

$$Cr\{[\sum_{m=1}^{20}\sum_{s=1}^{2} wic_m \times (1+\alpha) - \sum_{m=1}^{20}\sum_{s=1}^{2} Yic_m] + [\sum_{n=1}^{30}\sum_{s=1}^{2} wac_n \times (1+\beta) - \sum_{n=1}^{30}\sum_{s=1}^{2} Yac_n] + wmp_j \times (1+\eta) + wep_i\} \le \widetilde{q} \quad (2)$$

(2) Constraint of available water rights: Model (3) shows available water rights in the study region, which equals to available water resources minus water losses (due to evaporation and minimum ecological water requirements in the watercourse) and water rights for the municipality and ecology. In the study region, due to the policies associated with drinking safety and ecological security, water rights for the municipality and ecology should be guarded at first. Water rights for industry and agriculture should be adjusted in this IWRA model. Therein, $r$ is the minimum ecological water requirement in the watercourse (m$^3$); and $H$ is the evaporation of water resources (m$^3$):

$$Cr\{[\sum_{m=1}^{20}\sum_{s=1}^{2} wic_m \times (1+\alpha) - \sum_{m=1}^{20}\sum_{s=1}^{2} Yic_m] + [\sum_{n=1}^{30}\sum_{s=1}^{2} wac_n \times (1+\beta) - \sum_{n=1}^{30}\sum_{s=1}^{2} \times Yac_n]\}$$
$$\le \widetilde{q} - \widetilde{r} - H - [wmp_j \times (1+\eta) + wep_i] \quad (3)$$

(3) Constraint of ecological requirement: Models (4) and (5) display the minimum ecological requirements of the IWRA system, which are calculated by the Tenant method. Among them, T is the conversion coefficient, whose value can be regarded as $31.54 \times 10^6$; n is the statistical number of years; and $Q_{e\min}$ is the minimum monthly average runoff (m$^3$/s):

$$Cr\{\frac{T}{nn}\sum_{e=1}^{E} Q_{e\min}\} \le wep_i + \widetilde{r} \quad (4)$$

$$wep_i^{\min} \le wep_i \le wep_i^{\max} \quad (5)$$

Models (6) to (7) present the developing scale restrictions of population growth, agricultural expansion and industrial development, with consideration of various national strategies such as drinking safety and irrigative production security. In Model (8), the population growth should impose restrictions on allowable population scales. Meanwhile, water demand for industrial and agricultural uses should be restricted by the regional maximal allowance, as follows:

(4) Constraint of population growth and drinking safety:

$$0 \le wmp_j \le wmp_j^{\max} \quad (6)$$

(5) Constraint of agricultural development scale and irrigative production security:

$$0 \le Yac_n \le wac_n \le wac_n^{\max} \quad (7)$$

(6) Constraint of industrial development scale:

$$0 \le Yic_m \le wic_m \le wic_m^{\max} \quad (8)$$

### 2.3. Data Acquirement

In this study, the available water rights (i.e., $q$) are estimated by actual water availability and the regional water resources planning document. The net benefits of water resources are calculated by the regional statistics yearbook with consideration of economic development, which is displayed in Figure 4.

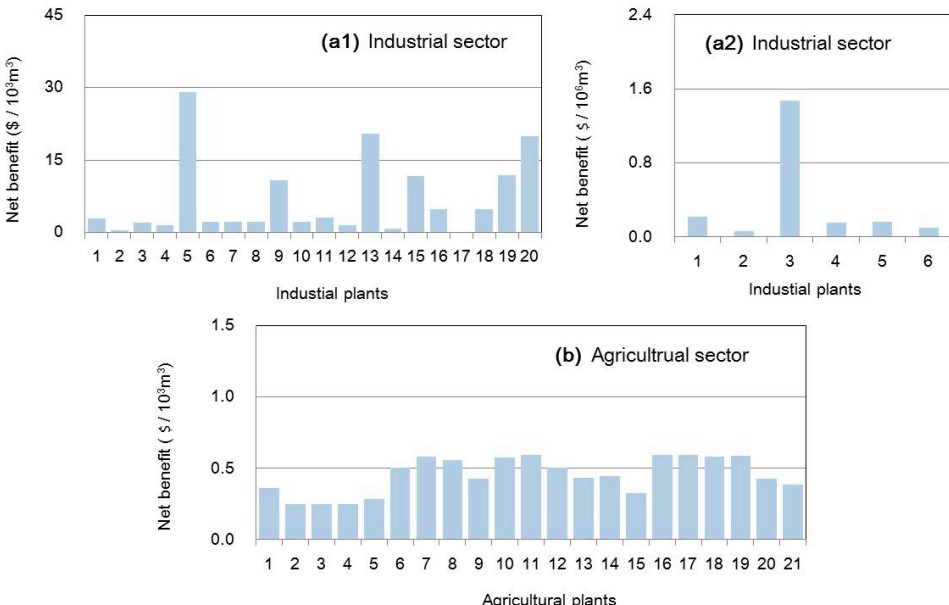

**Figure 4.** Net benefit among industrial and agricultural sectors. (figure (**a1**) presents net benefits in industrial sectors with lower values; figure (**a2**) presents net benefits in industrial sectors with higher values; figure (**b**) presents net benefits in agricultural sectors).

In fact, the total available water rights would be affected by natural and artificial factors. For example, precipitation can lead to fluctuations in available water resources, which may influence the available water rights. Meanwhile, artificial water resources management such as "three red lines" regulation may result in the reduction of total available water rights during planning periods. Thus, scenario assumption can be designed to reflect variations in available water rights, which is presented in Table 1. In this study, the "three red lines" requires the amount water usage to be decreased by 5% every five years in the future. Thus, six scenarios associated with water rights being withdrawn have been assumed based on the "three red lines" policy. Meanwhile, various techniques have been considered to increase the recycling ratio and water usage ratio in the municipality, industry and agriculture to address limited available water rights. Moreover, the risk attitude of the water manager can influence the scenario design. For example, a risk seeker would likely design a higher reduction of total water rights and the recycling/water usage ratio for a higher efficiency of water reuse, which may result in a higher benefit, but lead to a higher system failure risk. In contrast, a risk avoider would likely adopt a lower recycling ratio with consideration of the difficulty of popularizing technology. Thus, based on the SSLF method (as shown in the section on development of SSLF in an IWRA model), various scenarios can be designed as follows:

**Table 1.** Scenario design.

| Scenario | | | Assumption | | | |
|---|---|---|---|---|---|---|
| Scenario Sorting | Scenario Type | Probability of Scenario Occurrence | Reduction of Total Water Rights | Improvement of Water Recycling Ratio in Municipality | Improvement of Water Recycling Ratio in Industry | Improvement of Water Usage Ratio in Agriculture |
| S6 | RSS | 1/6 | 20% | 10% | 10% | 10% |
| S5 | RSS | 1/6 | 15% | 8% | 8% | 8% |
| S4 | NAS | 1/6 | 10% | 6% | 6% | 6% |
| S3 | NAS | 1/6 | 6% | 4% | 4% | 4% |
| S2 | RAS | 1/6 | 2% | 2% | 2% | 2% |
| S1 | RAS | 1/6 | 0% | 0% | 0% | 0% |

Note: risk seeking scenario denoted as "RSS"; neural attitudes scenario denoted as "NAS"; risk avoiding scenario denoted as "RAS".

## 3. Results

### 3.1. Water Rights Withdrawal

Figure 5 shows the solutions for total reduced initial water rights under various scenarios among the municipal, industrial, agricultural and ecological sectors when $\alpha$ levels are 0.6 and 0.99. The results show that reduction of initial water rights would change with the strict regulation of water resources. For instance, the highest reduction of initial water rights in industry and agriculture would be 64.42 and $7.13 \times 10^9$ m$^3$ under scenario 6, when the $\alpha$ level is 0.6. This indicates that the current irrigative water plan is irrational that should be adjusted, with the aim of improving the productivities of water resources. Meanwhile, since drinking safety and ecological security strategies have been advocated in northeastern China, water rights for the municipality and ecology can be safeguarded. Thus, there are no changes in the municipality and ecology under various scenarios. Moreover, the varied $\alpha$ levels can lead to different water rights being withdrawn, where higher $\alpha$ levels would lead to lower water-rights reductions, corresponding to a higher confidence level. For example, the reduction of initial water rights in agriculture would be 48.32 and $3.96 \times 10^7$ m$^3$ under scenario 5, when the $\alpha$ levels are 0.6 and 0.9.

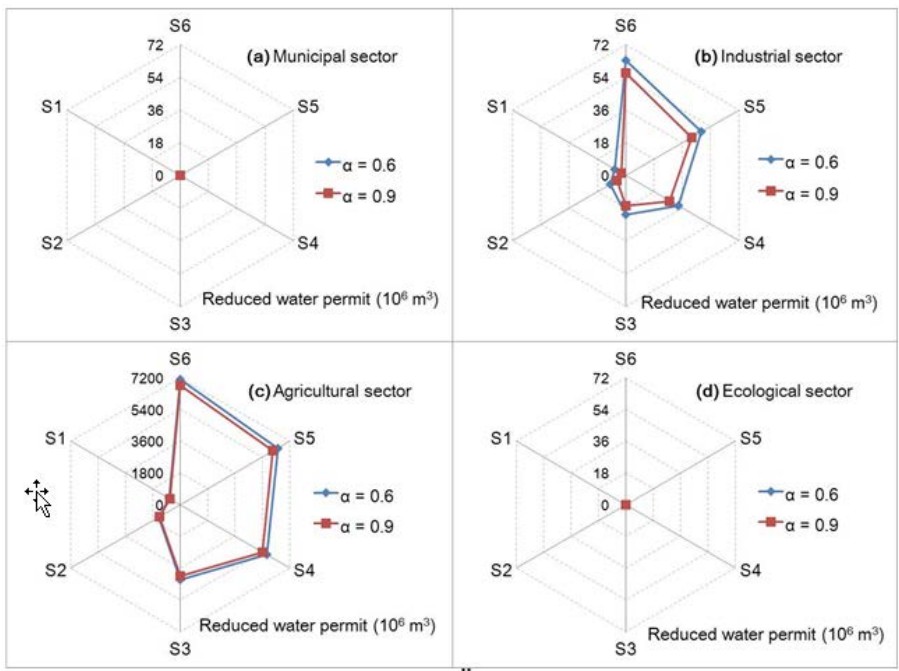

**Figure 5.** Reduced water rights among the municipal, industrial, agricultural and ecological sectors. (figure (**a**) represents reduced water right in municipal sector; figure (**b**) represents reduced water right in industrial sector; figure (**c**) represents reduced water right in agricultural sector; figure (**d**) represents reduced water right in ecological sector).

Figure 6 presents reduced initial water rights for agriculture from different sources (i.e., surface and underground water resources) under various scenarios when $\alpha$ levels are 0.6 and 0.99. The results show that the highest reduction of initial water rights ($3.24 \times 10^7$ m$^3$) would occur in agricultural plant 22 (i.e., irrigative district) under S1 (from surface water). In contrast, the lowest reduction of initial water rights would be 0, which indicates the number of initial water rights is suitable for the current level of irrigative development. In comparison, it was found that the initial water rights for agricultural plants 1, 2, 3, 4 and 5 would be from surface water; while the water rights for plants 9, 10, 11, 12, 13, 14 and 15 would be mainly from underground water. Moreover, the results show that the highest reductions from both surface and underground sources occur in agricultural plant 22. This indicated that, although plant 22 has the highest water rights value based on previous water intake permits, its lower productivity with the given water resources can lead to the highest water-rights withdrawal.

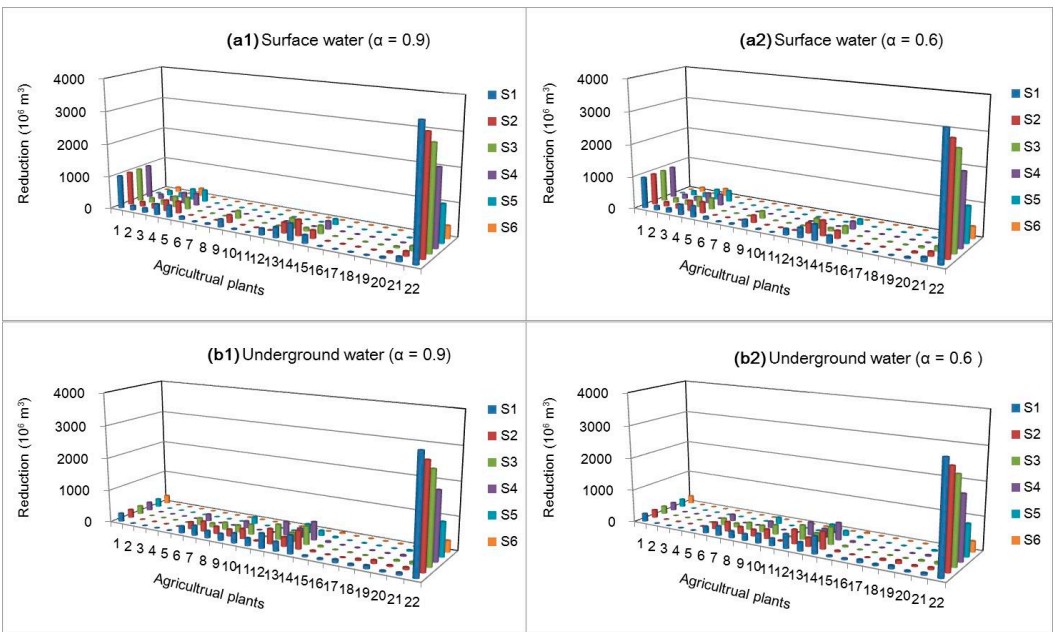

**Figure 6.** Reduced water rights from different sources for agricultural users under various scenarios. (figure (**a1**) presents reduced water right from surface water ($\alpha = 0.9$); figure (**a2**) presents reduced water right from surface water ($\alpha = 0.6$); figure (**b1**) presents reduced water right from underground water ($\alpha = 0.9$); figure (**b2**) presents reduced water right from underground water ($\alpha = 0.6$) )

*3.2. Optimal Water-Rights Allocation*

Figure 7 shows the optimized water-rights allocations for industrial users under different scenarios when $\alpha$ levels are 0.6 and 0.99. The results show that plant 15 can obtain the highest water-rights allocation ($1.53 \times 10^8$ m$^3$), as it has the highest productivity for water resources. In contrast, the lowest water-rights allocation ($0 \times 10^6$ m$^3$) occurs in plant 2, since it was closed down in 2010. Thus, its corresponding water rights should be reduced and transferred to other plants with high efficiency and productivity. Meanwhile, several $\alpha$ levels were analyzed based on the obtained expected demand, permit availabilities, and permit allocations under scenarios 1, 2, 4 and 6. The results show that a lower $\alpha$-level would lead to a higher water-rights allocation, but this corresponds to a higher risk violation (a lower credibility satisfaction level). The opposite results can be obtained when the $\alpha$-level is higher (i.e., $\alpha = 0.9$). For example, the water-rights allocations of plant 15 would be 8.76 and $8.29 \times 10^8$ m$^3$ under S1 when $\alpha$ is 0.6 and 0.9.

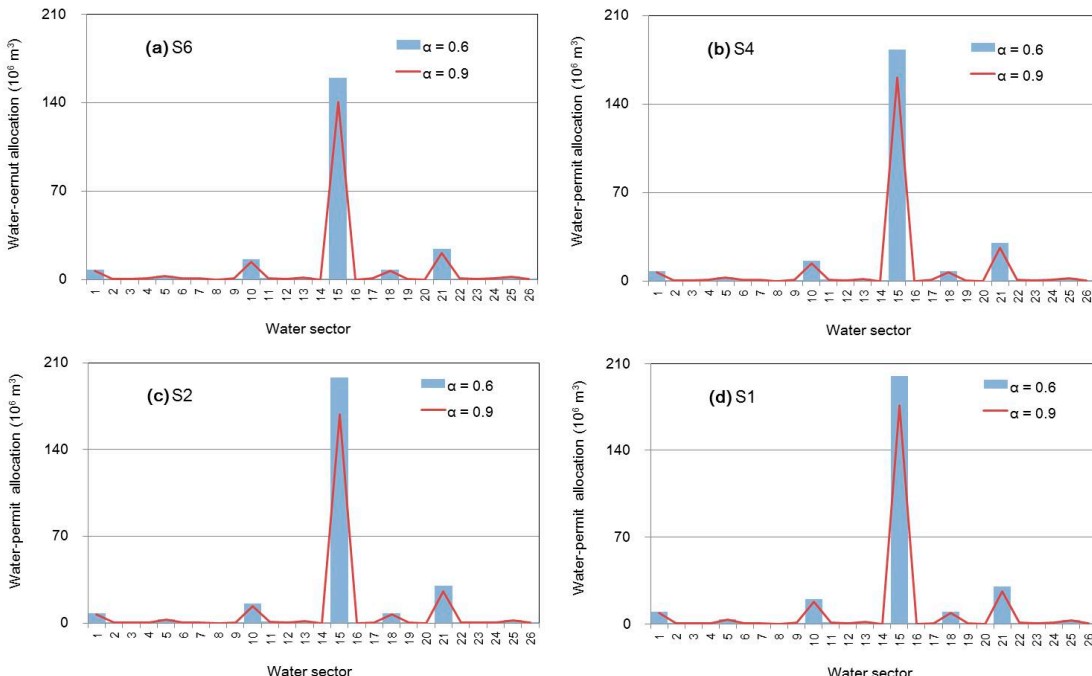

**Figure 7.** Optimal water-rights allocation for industrial users under various scenarios (figure (**a**) presents optimal water-right allocation under S6; figure (**b**) presents optimal water-right allocation under S4; figure (**c**) presents optimal water-right allocation under S2; figure (**d**) presents optimal water-right allocation under S1;)

Figure 8 displays the optimal proportion of water-rights allocation under various scenarios from surface and ground water ($\alpha = 0.9$). The results show that the highest optimized water-rights allocations would occur in agriculture (more than 90%), due the importance of grain production in the study region. In comparison, water rights for the municipality and industry are mainly from underground water. For instance, the proportion of water rights for municipal and industrial users would be 1.76% and 1.14% reserved for trading under S6, which may generate a higher system benefit if water rights can be traded in the market from underground water under S2. Inversely, the proportion of optimal water-rights allocation for agriculture is from surface water, which would be 94.92%, 94.71%, 94.66%, 94.55%, 94.49% and 94.44% under various scenarios.

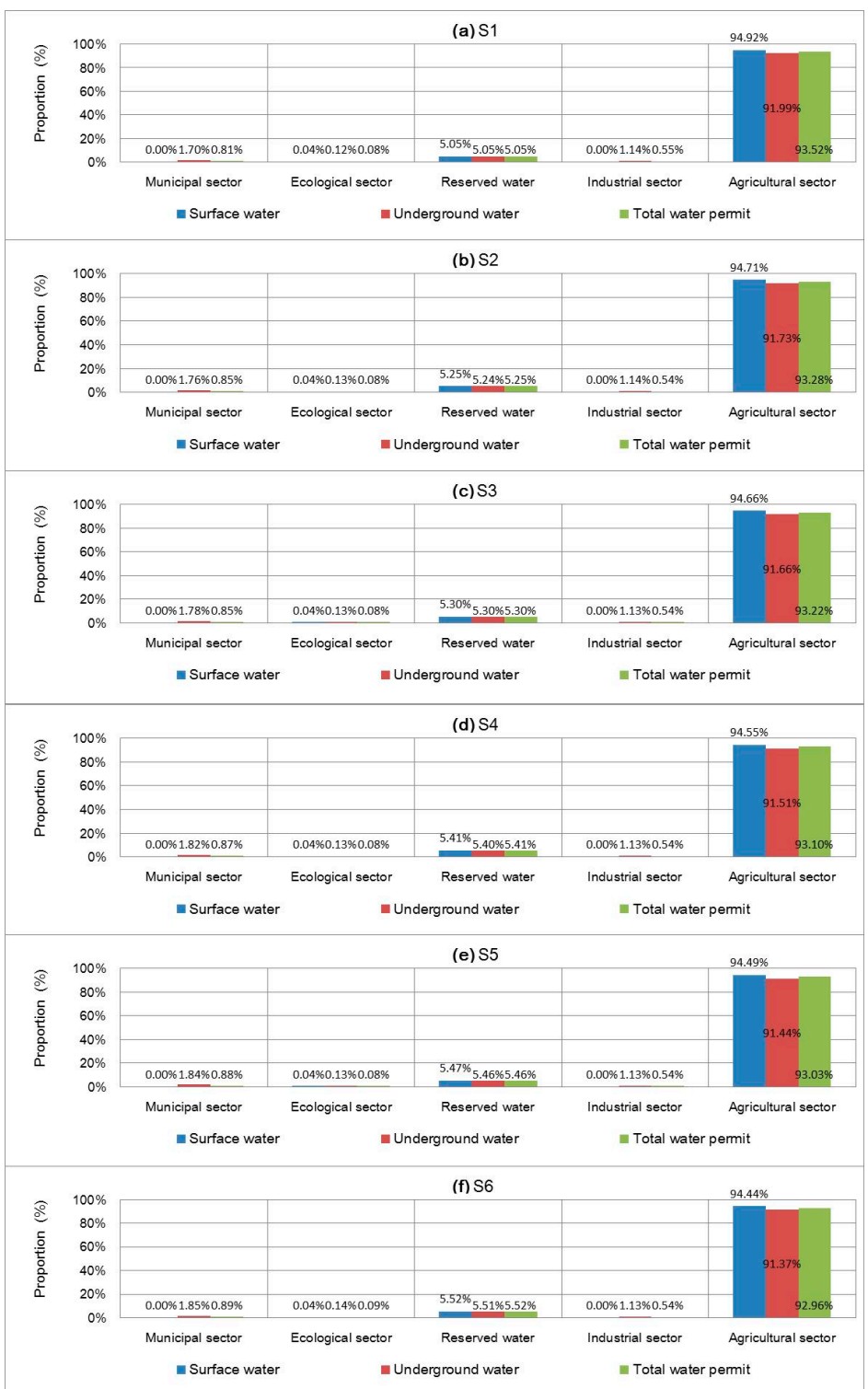

**Figure 8.** Optimal proportion of water-rights allocation from surface and ground water ($\alpha = 0.9$).

### 3.3. System Benefit

Figure 9 presents system benefits based on various initial water-rights allocations under various scenarios when $\alpha$ levels are 0.6, 0.7, 0.8 and 0.9. The lowest system benefits occur under scenario 6, where the values of the system benefits would be USD $0.90 \times 10^9$ ($\alpha = 0.6$) and USD $0.656.5 \times 10^9$ $\alpha = 0.9$). This implies that excessively restrictive regulation of water resources would result in economic losses due to water deficits. Compared to the system benefits of initial water-rights allocation based on

previous water intake permits, the system benefits under S1, S2, S3, S4 and S5 are higher than that allocation based on previous water intake permits.

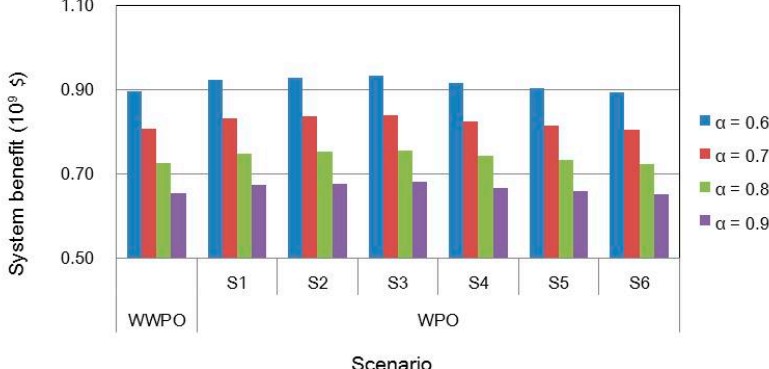

**Figure 9.** System benefits based on various initial water-rights allocations under various scenarios when $\alpha$ levels are 0.6, 0.7, 0.8 and 0.9 (WWPO denoted initial water-rights allocation based on previous water intake permits, WPO denoted as initial water-rights allocation with the IWRA model).

Figure 10 displays system benefits under S1, S3, S6 and Laplace criterion when $\alpha$ levels are varied. In this study, the scenarios can be divided into three types: risk seeking scenarios (S5 and S6), neutral attitude scenarios (S3 and S4), and risk avoiding scenarios (S1 and S2), as expressed in Table 1. The results demonstrate that S6 is associated with excessively restrictive water regulation (i.e., risk seeking scenario), and would maximize the reduction of initial water rights to produce extortionate losses due to water deficits, leading to lower system benefits. Under these situations, the water user cannot bear these losses, and would find techniques for promoting water productivity. In contrast, the opposite scenario (risk avoiding scenario) such as S1 can generate a lower water deficit, but may produce higher opportunity costs due to excessively higher than expected targets. In this case, it does not encourage water-saving and improvement of water productivity. Moreover, the Laplace scenario can produce the greatest system benefits in response to overall consideration of all risks of the scenarios, which can generate a more reliable and robust result.

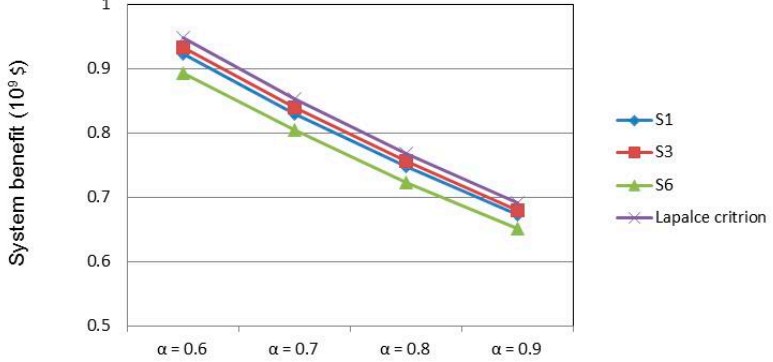

**Figure 10.** System benefit under S1, S3, S6 and Laplace criterion when $\alpha$ levels are varied.

## 4. Discussion

Figure 11 displays the optimal initial water-rights allocations and corresponding system benefits under various scenarios (associated with improvement of the water recycling ratio and water usage ratio) when the $\alpha$ level is 0.6. Since municipal and ecological water can be guarded, the improvement of the water recycling ratio and water usage ratio can play an important role in agricultural and industrial water-rights allocations. The results show that scenarios associated with a higher water recycling ratio (such as S6) could lead to a lower system benefit due to the investment of technique improvement

and losses relating to water-rights reduction. Under these situations, it may generate a lower system benefit in the short run. The opposite situation occurs in S1 (lowest improvement of water recycling ratio and water usage ratio), which can produce the highest benefit (USD $ 0.90 \times 10^9$) when the $\alpha$ level is 0.6. However, it is not beneficial for promoting water productivity in the long run.

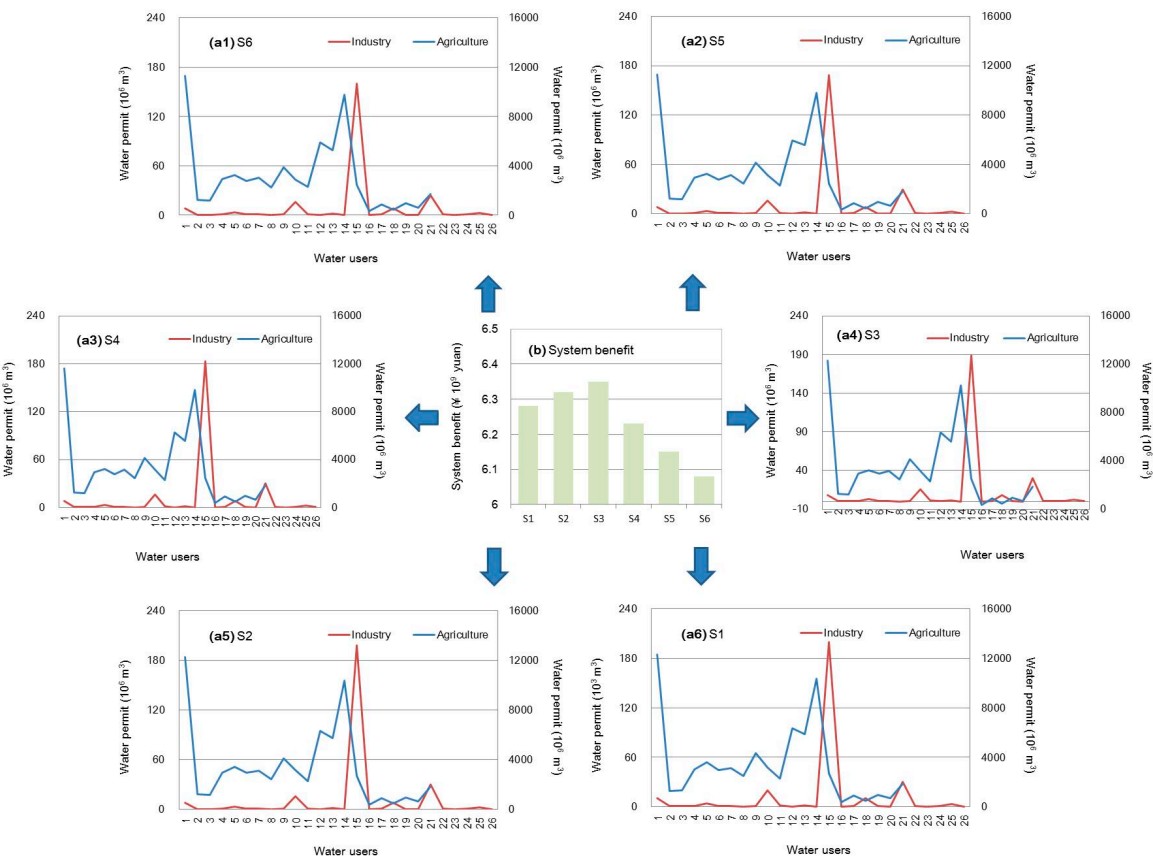

**Figure 11.** Optimal water-rights allocations and corresponding system benefits under various scenarios ($\alpha$ = 0.6).

## 5. Conclusions

In this study, an initial water-rights allocation (IWRA) model was built to replace the traditional water-rights allocation based on previous water intake permits, which can reallocate water rights to water users with higher efficiencies and productivities. The IWRA model can reflect the tradeoff between increasing water demands and limited available water rights under population growth and economic development. Meanwhile, stochastic scenario-based with Laplace criterion mixed fuzzy programming (SSLF) was proposed for the IWRA model to deal with uncertain information and interactions. SSLF can consider objective and subjective factors in a scenario analysis framework, which generates a set of 'possibility space' futures. Meanwhile, it can reflect risk attitudes of policymakers with the Laplace criterion, which can handle the probability of scenario occurrences under the supposition of no data being available, where the probability of scenario occurrence can be supposed to be reasonably equal. Moreover, SSLF is effective in tackling fuzziness expressed as probability distributions with high satisfaction degrees.

The proposed SSLF in the IWRA (SSLF-IWRA) is applied to a real case of water-rights allocation in Heilongjiang province, a typical alpine region in China. Results of the reduction of water rights, optimal initial water-rights allocation among varied industrial plants, and agriculture sectors from different water sources can be analyzed, which can be used to identify an optimized industrial structure and water resources management policies. It can also help policymakers with adjusting the current

industrial structure pattern and initial water-rights allocation schemes in a robust manner. Based on the regional situation of water resources, a number of scenarios associated with improvement of water recycling and water quantity regulation were designed and analyzed. The analysis shows that improvement of the water recycling ratio and water usage ratio could promote water productivities. Meanwhile, the results indicate that the "three red lines" policy can accelerate implementation of trading and water saving. Since available water rights will be reduced in the future five years due to the "three red lines" policy, water users would be expected to consider saving water or buying water from other sources, with the aim of reducing the economic losses due to water deficits. Based on an optimal initial water-rights allocation, a user with higher water productivity is more able to buy water from other sources or invest in water-saving technology.

Although the developed SSLF method can handle uncertain information within an IWRA and encourage water productivities in the study region, there are a number of limitations that should be considered. For instance, in a practical IWRA model, the varied uncertainties can lead to risks in water management. Therefore, an effective risk control method can be introduced into SFSL to enhance the robustness of the IWRA application. Meanwhile, in this IWRA model, the available water rights can reflect actual water availability dynamically, which is calculated by the mean value of water availability in recent years. Therefore, it cannot reflect recourse actions between expected water demand and available water rights. Correspondingly, more effective methods, such as two-stage stochastic programming, should be encouraged for adoption in future research.

**Author Contributions:** T.L. has constructed the idea of paper; X.Z. has designed the framework of SSLF and writing the paper; C.C. has revised the manuscript; X.K. has formulated the model; Y.Z. has designed scenario and made analysis; J.Z. has calculated the model and provide map; F.Z. has reproduced the obtained data and drawn the figures; H.D. has provided the various data in study region.

**Funding:** This research was funded by Self-designed Project of Heilongjiang Instituted of Water Conservancy Science (ZN201806); Beijing Social Science Fund (17LJC010); Open Research Fund Program form Fujian Engineering and Research Center of Rural Sewage Treatment and Water Safety (RST201810); the National Key Research Development Program of China (2018ZX07105); the National Natural Science Fund Project (41701621).

**Acknowledgments:** We are grateful for Si Zhengjiang and Zhang Shoujie providing policy and data support. We are deeply grateful to the reviewers for his/her insight and careful review.

**Conflicts of Interest:** There is no conflict of interest in this manuscript.

## Appendix A

Therefore, a scenario analysis (SA) method can be added to simplify the complex management issue into various scenarios that can reflect interactions between complex factors and decision outcomes, as follows [11,17]:

$$\max Outcome(A_h) = \max_{d \in D} S_{input}(a_h^1,\ a_h^2, \ldots,\ a_h^d) \tag{A1}$$

where $Outcome(A_h)$ is the decision outcome; $A_h$ is the outcome matrix row based on various scenario inputs ($A_h \in A, h = 1, 2, \ldots, H$); $S_{input}(a_h^1,\ a_h^2, \ldots,\ a_h^d)$ is varied scenario inputs, which can reflect numerous factors designed into scenario assumptions; h is the number of impact factors that can be considered in the scenario analysis; d and D are the options and corresponding option spaces, and $S_{input}$ is the overall performance. In an SA, the occurrence of various scenarios is random, thus stochastic programming can be introduced to express the probability of scenarios as probabilistic distributions, as follows [12,13]:

$$\max Outcome(A_{ih}) = \sum_{k=1}^{K} P_k * \max_{d \in D} S_{input}(a_{ih}^1,\ a_{ih}^2, \ldots,\ a_{ih}^d) \tag{A2}$$

where $P_i$ is the probability of each scenario occurrence; and k is the number of scenarios. In the process of initial water-rights allocation, various subjective estimations of policymakers such as risk attitudes can influence decision-making, resulting in political changes. In general, the risk attitude can be categorized as risk seeking, avoiding, and neutral attitudes, and can be an important

influence factor in scenario generation [7]. In a practical initial water-rights management issue, the risk attitude in a random scenario is difficult to calculate precisely. Meanwhile, the probability of scenario occurrence often appears as a random feature, which is influenced by the various private experiences and personality traits of policymakers. Therefore, a stochastic scenario analysis (SSA) can be introduced to reflect uncertain probabilities of scenario occurrence as the probability distribution. However, since the available data of the scenario associated with risk attitude is limited, SSA cannot be supported and regarded as a probability distribution. Therefore, Laplace's criterion (LC) is adopted for handling uncertain probabilities of scenario occurrence due to limited data availability [18]. In LC, it is supposed that the probabilities of scenario occurrence should appear reasonably equal if the sample size approaches infinity [13,18]. Under these assumptions, LC can help policymakers to reflect the balance between the expected payoff for each alternative (input performance) and choose alternatives with maximum values. Hence, a stochastic scenario analysis with Laplace's criterion (SSL) can be formulated as follows:

$$
\begin{aligned}
&\max Outcome_{Laplace}(A_{ih}) \\
&= \sum_{k=1}^{K} P_k \times \max_{d \in D} S_{input}(a_{ih}^1, a_{ih}^2, \ldots, a_{ih}^d) \\
&= \sum_{k=1}^{K} P_k \times \left\{ \max_{d \in D} S_{input} \left[ \begin{pmatrix} rsa_{i1}^1 & rsa_{i2}^1 & \ldots & rsa_{ih}^1 \\ rsa_{i1}^2 & rsa_{i2}^2 & \ldots & rsa_{ih}^2 \\ \ldots & \ldots & \ldots & \ldots \\ rsa_{i1}^d & rsa_{i2}^d & \ldots & rsa_{ih}^d \end{pmatrix} + \begin{pmatrix} raa_{i1}^1 & raa_{i2}^1 & \ldots & raa_{ih}^1 \\ raa_{i1}^2 & raa_{i2}^2 & \ldots & raa_{ih}^2 \\ \ldots & \ldots & \ldots & \ldots \\ raa_{i1}^d & psa_{i2}^d & \ldots & raa_{ih}^d \end{pmatrix} \right] \times (a_{ih}^1, a_{ih}^2, \ldots, a_{ih}^d) \right\} \\
&= \frac{1}{K} \times \left\{ \max_{d \in D} S_{input} \left[ \begin{pmatrix} rsa_{i1}^1 & rsa_{i2}^1 & \ldots & rsa_{ih}^1 \\ rsa_{i1}^2 & rsa_{i2}^2 & \ldots & rsa_{ih}^2 \\ \ldots & \ldots & \ldots & \ldots \\ rsa_{i1}^d & rsa_{i2}^d & \ldots & rsa_{ih}^d \end{pmatrix} + \begin{pmatrix} raa_{i1}^1 & raa_{i2}^1 & \ldots & raa_{ih}^1 \\ raa_{i1}^2 & raa_{i2}^2 & \ldots & raa_{ih}^2 \\ \ldots & \ldots & \ldots & \ldots \\ raa_{i1}^d & psa_{i2}^d & \ldots & raa_{ih}^d \end{pmatrix} \right] \times (a_{ih}^1, a_{ih}^2, \ldots, a_{ih}^d) \right\}
\end{aligned} \tag{A3}
$$

where $raa_{in}^d$ and $rsa_{in}^d$ are the scenarios with risk avoiding and seeking attitudes. Based on the SSL, the probability of a scenario can be supposed as 1/K. However, in an initial water-rights management issue, some parameters (such as limited economic data or meteorological data) in the right- and left-hand sides of objective functions and constraints are expressed as vagueness due to limited data and estimative error, which cannot be tackled by SSL. Hence, a type of fuzzy programming called fuzzy credibility constrained programming (FCP) can be added to reflect fuzzy information regarded as the possibility distribution, which can express the relationship between the satisfaction degree and system-failure risk as follows [29,30]:

$$
\begin{aligned}
&\max Outcome_{Laplace}(A_{ih}) \\
&= \frac{1}{K} \times \left\{ \max_{d \in D} S_{input} \left[ \begin{pmatrix} rsa_{i1}^1 & rsa_{i2}^1 & \ldots & rsa_{ih}^1 \\ rsa_{i1}^2 & rsa_{i2}^2 & \ldots & rsa_{ih}^2 \\ \ldots & \ldots & \ldots & \ldots \\ rsa_{i1}^d & rsa_{i2}^d & \ldots & rsa_{ih}^d \end{pmatrix} + \begin{pmatrix} raa_{i1}^1 & raa_{i2}^1 & \ldots & raa_{ih}^1 \\ raa_{i1}^2 & raa_{i2}^2 & \ldots & raa_{ih}^2 \\ \ldots & \ldots & \ldots & \ldots \\ raa_{i1}^d & psa_{i2}^d & \ldots & raa_{ih}^d \end{pmatrix} \right] \times (a_{ih}^1, a_{ih}^2, \ldots, a_{ih}^d) \right\}
\end{aligned} \tag{A4}
$$

subject to

$$
Cr \left\{ \sum_{m=1}^{M} u_{mn} w_m \leq \widetilde{c}_n \right\} \geq \lambda, n = 1, 2, \ldots, M; n = 1, 2, \ldots, N \tag{A5}
$$

$$
w_m \geq 0, m = 1, 2, \ldots, M \tag{A6}
$$

where $w = (w_1, w_2, \ldots, w_m)$ is a vector of non-fuzzy decision variables, and m and n are subscripts of constraint. $u_{mn}$ and $c_n$ are cost, technical, and right-hand side coefficients. Among these parameters, the right-hand side coefficient $c_n$ is represented as fuzzy sets. In order to improve the quality of fuzzy expression, the credibility measure is introduced to express fuzzy sets based on the concept of possibility, necessity, and credibility. In general, two basic types of fuzzy expression (possibility, necessity) are often used for reflecting risk violations and confidence degrees, which can be expressed as

follows: $Pos\{\varepsilon \le s\} = \sup_{u \le s} \mu(u)$, $Nec\{\varepsilon \le s\} = 1 - \sup_{u > s} \mu(u)$, where $\varepsilon$ is a fuzzy variable with membership function $\mu$, and let $\mu$ and r be real numbers. Then, a credibility measure can couple the possibility and necessity measures into a framework to improve the quality of fuzzy expression, which can be represented as $Cr\{\varepsilon \le s\} = 0.5 \times (Pos\{\varepsilon \le s\} + Nec\{\varepsilon \le s\})$ [22,31]. In general, the value of the credibility level should be greater than 0.5 in response to avoiding improper outcomes and violated risks [20,30]. Under these situations, the constraint (4b) can be a proven credibility measure when $\lambda > 0.5$, as follows:

$$\sum_{m=1}^{M} u_{mn} w_m \le c_n^2 + (1 - 2\lambda)(c_n^2 - c_n^1), n = 1, 2, \ldots, N_1 \tag{A7}$$

Therefore, stochastic scenario-based with Laplace criterion mixed fuzzy programming (SSLF) can be formulated as follows:

$$\max Outcome_{Laplace}(A_{ih})$$

$$= \frac{1}{K} * \left\{ \max_{d \in D} S_{input} \left[ \begin{pmatrix} rsa_{i1}^1 & rsa_{i2}^1 & \cdots & rsa_{ih}^1 \\ rsa_{i1}^2 & rsa_{i2}^2 & \cdots & rsa_{ih}^2 \\ \cdots & \cdots & \cdots & \cdots \\ rsa_{i1}^d & rsa_{i2}^d & \cdots & rsa_{ih}^d \end{pmatrix} + \begin{pmatrix} raa_{i1}^1 & raa_{i2}^1 & \cdots & raa_{ih}^1 \\ raa_{i1}^2 & raa_{i2}^2 & \cdots & raa_{ih}^2 \\ \cdots & \cdots & \cdots & \cdots \\ raa_{i1}^d & psa_{i2}^d & \cdots & raa_{ih}^d \end{pmatrix} \right] * (a_{ih}^1, a_{ih}^2, \ldots, a_{ih}^d) \right\} \tag{A8}$$

subject to

$$\sum_{m=1}^{M} u_{mn} w_m \le c_n^2 + (1 - 2\lambda)(c_n^2 - c_n^1), n = 1, 2, \ldots, N_1 \tag{A9}$$

$$w_m \ge 0, m = 1, 2, \ldots, M \tag{A10}$$

## Appendix B

| | |
|---|---|
| $Outcome_{Laplace}(A_{ih})$ | the objective function under Laplace criterion ($) |
| $A_{ih}$ | the outcome matrix row based on various scenario input ($) |
| k | the number of scenarios |
| $rsa_{ih}^d$ | the different inputs under risk-seeking scenario |
| $raa_{ih}^d$ | the different inputs under risk-avoiding scenario |
| h | the number of impact factors that can be considered into scenario analysis |
| d and D | the options and corresponding option spaces |
| m | the number of water users in industry |
| n | the number of water users in agriculture |
| s | the different water sources; s = 1 is ground water source; s = 2 is underground water |
| $BIC_{ms}$ | the net income when per unit water rights being delivered to satisfy the expected target in industry ($/m$^3$) |
| $wic_{ms}$ | the initial water right empowerment based on previous water intake permit in industry (m$^3$) |
| $BAC_{ns}$ | the net income when per unit water rights being delivered to satisfy the expected target in industry ($/m$^3$) |
| $wac_n$ | the initial water right empowerment based on previous water intake permit in in agriculture (m$^3$) |
| $BMC_j$ | the net income when per unit water rights being delivered to satisfy the expected target in municipality ($/m$^3$) |
| $wmp_j$ | the initial water right empowerment based on previous water intake permit in in municipality (m$^3$) |
| $BEC_i$ | the net income when per unit water rights being delivered to satisfy the expected target in ecology ($/m$^3$) |
| $wep_i$ | the initial water right empowerment based on previous water intake permit in in ecology (m$^3$) |
| $Yic_{ms}$ | the amount of water-rights reduction in industry (m$^3$) |
| $Yac_{ns}$ | the amount of water-rights reduction in agriculture (m$^3$) |
| $TMC_j$ | the amount of water-rights reduction in municipality (m$^3$) |
| $CIC_{ms}$ | the economic losses due to insufficient water rights brought to an industrial user after water-rights reduction / withdrawn ($/m$^3$) |

| $CAC_{ns}$ | the economic losses due to insufficient water rights brought to an agricultural user after water-rights reduction / withdrawn ($/m$^3$) |
|---|---|
| $TIC_{ms}$ | the cost of improvement of water recycling ratio in industry ($/m$^3$) |
| $TAC_{ns}$ | the cost of improvement of water usage ratio in agriculture ($/m$^3$) |
| $\alpha$ | the improvement of water recycling ratio in industry |
| $\beta$ | the improvement of water usage ratio in agriculture |
| $\eta$ | the improvement of water recycling ratio in municipality |
| $q$ | the total water availability in study region (m$^3$) |
| $r$ | the minimum ecological water requirement in the watercourse (m$^3$); |
| $H$ | the evaporation of water resources (m$^3$) |
| T | conversion coefficient, |
| $Q_{e\min}$ | minimum monthly average runoff (m$^3$/s) |
| nn | statistical number of year |
| $wep_i^{\min}$ | minimum ecological water requirement (m$^3$) |
| $wep_i^{\max}$ | maximum ecological water requirement (m$^3$) |
| $wmp_j^{\max}$ | maximum water requirement in municipality with consideration population growth (m$^3$) |
| $wac_n^{\max}$ | maximum water requirement in agriculture with consideration economic development (m$^3$) |
| $wic_m^{\max}$ | maximum water requirement in industry with consideration economic development (m$^3$) |

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
