# Peer review of "Scenario Analysis of Initial Water-Rights Allocation to Improve Regional Water Productivities"

_water, doi:10.3390/w11061312_

Round 1

Reviewer 1 Report

1.     It is suggested not to use the word “optimal” in the terms such as “Identifying optimal water-allocation alternatives”. Optimal is a strong word. The results are obtained from your optimization model, but there is no guarantee that they are absolutely “optimal”.

2.     The abstract should be revised where the main findings and novelty should be made clear.

3.     This paper could be better organized. for example, section 2.2 is too long and could be divided into two sections.

4.     Pages 8 to 10, more details about the model are needed. For example: Who is the manager and what are his/her goals? How to get the expected target from various water users (or sectors)? How did the manager allocate the water permit? What is the relationship between intake water permit and tradable water permit? How often does manager make a plan?

5.     Please provide more details about the fuzzy credibility programming technique used in Model 7. Recent studies on credibility constrained programming should cited:

a.      Li, Zhong, et al. "Inexact two-stage stochastic credibility constrained programming for water quality management." Resources, Conservation and Recycling 73 (2013): 122-132.

b.     Xu, Jiaxuan, et al. "A two-stage fuzzy chance-constrained water management model." Environmental Science and Pollution Research 24.13 (2017): 12437-12454.

6.     Please consider providing a step-by-step description of the solution algorithm that was used to solve the SSLF-WPA model.

7.     What are the assumptions of the scenarios in Table 2? Why the scenarios are designed in this particular way?

8.     Disadvantages of the proposed model should be provided in the Conclusion section. Suggestions for future studies should also be provided.

9.     This paper should be proofread by a native English speaker. There are a number of typographical and grammatical errors that need to be corrected.

Author Response

May 23th, 2019

Dear reviewer,

We much appreciate your and reviewers' insightful comments and suggestions, and have revised the paper accordingly. Attached please find:

(1) revised manuscript, and

(2) responses to the reviewer’s comments

Thank you very much again for your time and kind consideration. We look forward to your response.

Sincerely yours,

X.T. Zeng, Ph.D.

Reviewer 2 Report

This study might be a very good modeling exercise and could have some scientific significance in managing water right and water transfer, but the authors used non-standard English and confusing sentences, which made it very hard to understand what the authors wanted to say or to judge the overall merit of the study. 

What is a “water-permit allocation”? Is it water allocation? If it is, please simply say it. Water permits are not allocated. They are issued in a certain water right system that governs the terms and conditions associated with them.

Please explain what “initial water-permit confirmation” is. The term is not defined in the manuscript and is not a commonly understood word in water rights.

It is not clear to me what the problem statement of the manuscript is by reading the manuscript. The authors said water-permit allocation was irrational. But how is it irrational? What is the current water right system in the alpine region of China? Why such a system could reduce efficiency? How is “efficiency” defined here? How the proposed WPA model can address or help address this problem?   

Since the problem that authors were trying to address is not known, it is also not clear to me how the previous studies that the authors cited are related to the authors’ objective or how those cited studies help the authors developed their own study.

Please consider putting the detailed formulae of the objective functions and their explanation in an appendix as appropriately. The method section should focus on a clear but brief introduction of the method, a summary of the programming and the principle behind them.   

Specific examples of confusing wording are as follow. There are many others in the manuscript. I encourage the authors resubmit the manuscript after correcting and clarifying them.

Line 19 and Line 54: What exactly does WMTP stand for? The phrase immediately before it is “tradable water permits mechanism”. So it should be shorthanded as TWPM, shouldn’t it?

19: Suggest changing “confront” to “address”

Line 22: What does “conclude” actually mean here? Do you mean “include”?

Line 28: There is an “is” between “it” and “found”

Line 30: When you say “tradeoff”, do you mean “transfer”?

Line 49: “perused” does not seem to be a right word here. It means read something thoroughly or carefully.

Author Response

(The authors gave the same response as above.)

Reviewer 3 Report

Dear Authors,

An important issue was discussed, i.e., water allocation which is one of the fundamental of water resources management. However, the style of the language makes so hard to follow through. Anyway, it was reviewed and some important issues were raised as can be found attached herewith which I think can improve the quality of the manuscript.

Thank you

Author Response

(The authors gave the same response as above.)

Round 2

Reviewer 3 Report

Dear Authors,

I recommend the effort you put in reviewing this manuscript which I am sure will be interested to scientific community. However, there are still some issues with regard to English, which i hope you will look at it again  in order to communicte effectively. Attached i made some comments.

Thank you.

Author Response

We want to express our deep gratitude to reviewer for his careful and helpful comments. Accordingly, we have made revision in the attachment. 
